# Study on the Adhesion Performance of Asphalt-Calcium Silicate Hydrate Gel Interface in Semi-Flexible Pavement Materials Based on Molecular Dynamics

**DOI:** 10.3390/ma14164406

**Published:** 2021-08-06

**Authors:** Bei Hu, Wenke Huang, Jinlou Yu, Zhicheng Xiao, Kuanghuai Wu

**Affiliations:** School of Civil Engineering, Guangzhou University, Guangzhou 510006, China; 13026335960@163.com (B.H.); 18711751239@163.com (J.Y.); xzc297055783@163.com (Z.X.); wukuanghuai@163.com (K.W.)

**Keywords:** asphalt, C-S-H gel, interface, molecular dynamics, adhesion performance

## Abstract

The interface between an asphalt binder and a calcium silicate hydrate (C-S-H) gel is a weak point of semi-flexible pavement material. In this study, the adhesion performance of asphalt-C-S-H gel interface in semi-flexible pavements at a molecular scale has been investigated. Molecular dynamics (MD) simulations were applied to establish three asphalt binders: 70# asphalt binder (the penetration is 70 mm), PG76-22 modified asphalt binder (a kind of asphalt binder that can adapt to the highest temperature of 76 °C and the lowest temperature of −22 °C), and S-HV asphalt binder (super high viscosity). The effects of different temperatures and SBS modifier contents on interfacial adhesion were explored. The obtained results showed that temperature variations had little effect on the adhesion work of the asphalt-C-S-H gel interface. It was also found that by increasing the content of SBS modifier, the adhesion work of the asphalt-C-S-H gel interface was increased. The molecular weight of each component was found to be an important factor affecting its molecular diffusion rate. The addition of SBS modifier could regulate the adsorption of aromatics by C-S-H gel in the four components of asphalt binder and improve the adsorption of resins by C-S-H gel.

## 1. Introduction

Traditional asphalt pavements are widely used in road engineering due to the good adhesion between asphalt binder and aggregates. However, asphalt pavement, as a viscoelastic material [1,2,3] suffers from long-term aging, rutting, fatigue cracking, and water erosion, which seriously affect its durability [4,5,6,7,8]. Compared with traditional asphalt pavement, semi-flexible pavement (SFP) has remarkable stiffness and durability and excellent pavement bearing capacity and rutting resistance [9,10,11,12,13,14,15].

In previous research, great efforts have mainly been made to improve the performance of asphalt binders, aggregates, and cement mortar in semi-flexible pavements, as well as their influence on cracked materials [16]. Yang [17] performed a cyclic wheel load and runway test-bed tests and found that the mechanical characteristics of semi-flexible materials were improved when the asphalt mixture air void was up to 26%. Zhang et al. [18] studied the effects of the compositions and formulations of cement paste and cement mortar on grouting performance, compared them in terms of fluidity, strength, and dry shrinkage, and finally found the best formula: the best water–cement ratio is 0.58, the best fly ash content is 10%, and the best mineral powder content is 10%. Ding et al. [19] found that the high- and low-temperature stabilities of semi-flexible pavement materials were higher than those of ordinary asphalt pavement materials and an increase in porosity improved the performance of matrix asphalt mixtures. Setyawa [20] studied the compressive performance of grouted macadam and determined the relationships between asphalt skeleton, cement slurry, and the type and size of aggregate. Husain [21] investigated the aggregate gradation, durability, and strength of cementitious materials through statistical analyses and obtained the effect of gradation on SFP material properties.

Although a great number of research has been conducted on the macroscopic performance of semi-flexible pavements, the asphalt-C-S-H gel interface is a weak point of this kind of pavement material and only microscopic analyses can reveal the interaction mechanism of the asphalt-C-S-H gel interface. As an increasingly mature tool, molecular dynamics (MD) can not only directly simulate the macroscopic evolution of matters, but also provide clear images of microstructures and particle motion, as well as their relationships and macroscopic properties. The application of MD simulation in asphalt mixtures is becoming increasingly popular, as it can describe the behavior of asphalt at a molecular level and determine the relationship between chemical composition and asphalt binder macroscopic properties such as viscosity, aging effect, self-healing performance, and diffusion behavior. These findings can also be applied to study the nano-mechanical properties of asphalt binders and asphalt aging [22,23,24,25,26,27].

In this study, MD was applied to establish 70# asphalt-C-S-H gel, PG76-22 modified asphalt-C-S-H gel, and S-HV modified asphalt-C-S-H gel models. In terms of the three aspects of adsorption strength, diffusion law and solubility law, it was found that the adhesion ability of the asphalt-C-S-H gel interface with different SBS contents was affected by variations in temperature and asphalt binder composition.

## 2. Materials and Methods

### 2.1. Materials

#### 2.1.1. Asphalt Binder Model

Asphalt has very complex chemical compositions with a variety of hydrocarbons and their non-metallic derivatives (a mixture of hydrocarbons and non-hydrocarbons). Therefore, it cannot be described as a single pure compound. In 1969, Corbett [28] developed a four-component analysis method to classify asphalt into saturates, aromatics, resins, and asphaltenes, which were widely applied in the design and micro-mechanism of asphalt materials on the molecular scale. Using this four-component asphalt model, researchers have evaluated the compatibility of asphalt and modifiers, the adhesion between asphalt and aggregate, and the effect of oxidative aging on asphalt binders [29,30,31,32]. In this research, a four-component asphalt model was established based on MS molecular dynamics software to perform simulation experiments, as shown in Figure 1.

In order to obtain a reasonable four-component asphalt model, initial density was set to 1 g/cm^3^ and pre-equilibrium operation of 100ps was carried out under NPT ensemble (temperature and pressure were set to 298K and 1 atm, respectively). Then, NPT simulations were performed under 500ps to reduce the unreasonable structures in the model. Finally, a stable asphalt model was obtained by 100ps simulation in NVE ensemble. The optimized asphalt model is shown in Figure 2.

In order to investigate the influence of different asphalt types on the interface, styrene–butadiene–styrene (SBS) was selected as an asphalt modifier and its molecular formula is shown in Figure 3. Asphalt with 4.27% SBS modifier was adopted for the simulation of PG76-22 modified asphalt and 11.8% SBS modifier was used for ultra-high viscosity modified asphalt (S-HV). The optimized molecular models of the two modified bitumen types are presented in Figure 4. The detailed compositions of the three asphalt binders are listed in Table 1.

#### 2.1.2. C-S-H Gel Model

The hydration product of ordinary portland cement is mainly calcium silicate hydrate (C-S-H), which is the key component determining cement properties. Figure 5 presents the SEM image of C-S-H gel. Since Grudemo [33] and Taylor [34] first studied the hydration products of cement, many researchers have carried out in-depth research on these products, developed a series of molecular models, and applied them in practical engineering. At present, the models developed by Hamid [35] and Bonaccorsi [36] are considered the best C-S-H models. In this study, the Bonaccorsi model was adopted as the initial C-S-H gel model in MD simulations.

The chemical formula of C-S-H is Ca_9_Si_6_O_18_(OH)_6_·8H_2_O. In this study, the component proportions given in Table 2 were used to prepare C-S-H gel unit cells using MS software. The molecular model of each component is shown in Figure 6. The edge lengths a_m_, b_m_, and c_m_ of the unit cells were 1.0576nm, 0.7265nm, and 1.0931nm, respectively. In order to make simulation systems more realistic, it was necessary to expand single cells. In this research, the dimensions of expanded crystals were 4a_m_ × 3b_m_ × 2c_m_.

In order to establish an optimal C-S-H gel model, first the developed model was optimized. The initial density of gel model in this study was set to 2.325 g/cm^3^. 100 ps and 50 ps simulations were carried out under NPT and NVE ensembles, respectively (the temperature was set to 298K and pressure was 1atm). Then, an NPT calculation was performed under 500ps condition to reduce the number of unreasonable structures in the model. Finally, the optimal model was obtained by 100ps calculation in an NVE ensemble. The C-S-H gel molecular structure is shown in Figure 7.

#### 2.1.3. Asphalt–C-S-H gel Model

In order to study the adhesion properties of different types of asphalt components on C-S-H gel surface at different temperatures, interfacial molecular models were constructed for different asphalt types and C-S-H gel. The models had a three-layer structure with the lower layer being C-S-H gel mixture, the middle layer being asphalt mixture, and the upper layer being the 50 Å vacuum layer. Four temperatures of −15 °C, 25 °C, 65 °C, and 105 °C were simulated. The interface models, as shown in Figure 8, were named Model A (70#asphalt), Model B (PG76-22 modified Asphalt), and Model C (S-HV modified asphalt).

### 2.2. Evaluation Index

#### 2.2.1. Adhesion Work

Adhesion work between asphalt and C-S-H gel is defined as the decrease of interfacial free energy of the whole system due to the adhesion of asphalt to C-S-H gel. That is, increase of adhesion work increased the amount of released energy and improved adhesion performance. In this work, adhesion work was applied for the characterization of adsorption strength between different types of asphalt and C-S-H gel, as expressed in Equation (1).
△*E* = *E_a_* + *E_b_* − *E_ab_*,(1)
where △*E* is adhesion work between asphalt and C-S-H gel (kcal/mol); *E_a_* is energy in the presence of asphalt alone (kcal/mol); *E_b_* is energy in the presence of C-S-H gel alone (kcal/mol); and *E_ab_* is the total energy of asphalt and C-S-H gel (kcal/mol).

Among them, adhesion was composed of Van der Waals and Coulomb electrostatic forces, as stated in Equation (2).
△*E* = △*E_vdw_* + △*E_coulomb_*,(2)
where △*E_vdw_* and △*E_coulomb_* are adhesive works due to Van der Waals and Coulomb electrostatic forces (kcal/mol), respectively.

#### 2.2.2. Diffusion Coefficient

The diffusion coefficient is an important index in measuring the permeation and diffusion abilities of particles. In this study, the diffusion coefficient was applied for the characterization of the movement and migration abilities of the four components of asphalt on C-S-H gel surface. Using the Einstein equation, the diffusion coefficient was calculated from the long-time limit of mean azimuth shift. Diffusion coefficient and mean square displacement (MSD) were calculated using Equations (3) and (4), respectively.
(3)$$MSD\left(t\right)=\langle {\left|r_{i}\;(t)-r_{i}\;(0)\right|}^{2}\rangle ,$$
where *r_i_* (t) is the position vector of particle *i* at time t; *r_i_* (0) is the position vector of particle *i* at initial time; and < > denotes the average of all atoms in the group.
(4)$$D=\frac{1}{6}\underset{t\to \infty }{\lim}\frac{d}{dt}\sum_{i=1}^{N}\langle {|r}_{i}{\;(t)-r}_{i}{\;(0)|}^{2}\rangle ,$$
where *N* is the number of particles diffused in the system and *D* is the diffusion coefficient of particles (m^2^/s).

#### 2.2.3. Relative Concentration

Relative concentration can characterize the aggregation degree of substances in a certain range. It was found that the increase in relative concentration increased the content of asphalt components away from the surface of the C-S-H gel. When an asphalt component had a continuous peak concentration in a range away from the surface of the C-S-H gel, that asphalt component was aggregated in that range. The relative concentration profile of each asphalt component was applied to characterize its concentration variation on C-S-H gel surface at different temperatures, as stated in Equation (5).
(5)$$\Omega =\frac{m_{i}}{M_{I}}\;\times \;100\%,$$
where ω is relative concentration; I is the mass of an asphalt component in a layer; i represents any of the four components of asphalt, i.e., saturates, aromatics, resins, and asphaltenes; and $M_{i}$ is the total mass of an asphalt component.

### 2.3. Methods

The research method and technical workflow of this paper are shown in Figure 9. Firstly, three asphalt models and the C-S-H gel model were established, and then the established models were verified. Finally, the results were compared and analyzed based on three evaluation indexes.

## 3. Results

### 3.1. Model Verification

#### 3.1.1. Asphalt Model

In order to verify the validity and applicability of the proposed asphalt molecular model, density and solubility parameters were adopted as evaluation indicators. The simulation and experimental results are summarized in Table 3.

It can be seen from Table 3 that the density and solubility of the asphalt structure simulated in this paper were both within the range of practical reference values. Among them, the simulation value of relative density was close to the lower limit, while that of the solubility parameter was close to the upper limit of the actual reference value. Due to differences in the composition of material complexity, a difference between simulated and actual values in this range was acceptable. In terms of relative density and solubility parameters, the asphalt molecular model developed in this study was found to be precise and reliable and could be used in subsequent MD simulations.

#### 3.1.2. C-S-H Gel Model

In order to verify the validity of the C-S-H gel model developed in this research, its physical properties after optimization were simulated. According to calculations, the Young’s modulus of the gel model along the X direction was 31.6 GPa, which was consistent with the Young’s modulus range of C-S-H gel reported in references [38,39,40,41,42]. The calculated Young’s modulus was close to the upper limit of Young’s modulus range reported in the literature, because there was a certain difference between the purity of real and simulated materials and crystal was prone to defects. However, the physical values calculated by simulation were still within the range of experimentally obtained values. Therefore, the C-S-H gel model established in this study proved to be more reasonable and could be applied for subsequent MD simulations.

### 3.2. Adsorption Strength

According to the results obtained from MD calculations, using Equations (1) and (2) and the principle of adsorption strength, the values of the three types of asphalt and C-S-H gel at different temperatures were calculated, as given in Table 4.

It can be seen from Table 4 that for different types of asphalt, adhesion work between asphalt and C-S-H gel was mainly due to Coulomb electrostatic force, which accounted for more than 99%. The reason for this observation is that C-S-H gel is alkaline and has electrostatic interactions with polar molecules in asphalt. For the same type of asphalt, slight adhesion energy changes were witnessed at different temperatures, which indicated that temperature had little effect on the adhesion work between asphalt and C-S-H gel. Regarding different bitumen types at the same temperature, the order of adhesion work was 70# asphalt < PG76-22 modified asphalt < S-HV modified asphalt, which was because the SBS modifier content affected adhesion such that higher SBS modifier contents resulted in greater adhesion between the asphalt and C-S-H gel.

### 3.3. Diffusion Law

In order to analyze the diffusion laws of different asphalt types at different temperatures, as shown in Figure 10, the diffusivity changes of each component of 70# asphalt, PG76-22 modified asphalt, and S-HV modified asphalt at −15 °C, 25 °C, 65 °C, and 105 °C.

As can be seen from Figure 10a, the order of the diffusivity of the four components of asphalt at the same temperature was asphaltenes < resins < aromatics < saturates. This was contrary to the order of the molecular weight of asphalt components. Among them, the diffusion rate of saturates were found to be the highest and most affected by temperature, especially at temperatures above 65 °C. This was mainly because the diffusion rates of the four components of asphalt were strongly and negatively related to their molecular weight.

As could be seen from Figure 10b, the diffusion rate of light component saturates were the highest, followed by the aromatics component. Compared with 70# and S-HV modified asphalt types, the diffusion rate of saturates in PG76-22 modified asphalt was linearly increased. This revealed that the diffusion rate of saturates could be steadily increased by adding appropriate amounts of SBS modifier. At temperatures below 20 °C, the diffusion rate of asphaltenes was higher than that of resins. However, at temperatures above 20 °C, the diffusion curves of asphaltenes and resins were intertwined with each other, indicating that SBS modifier could enhance the diffusion rate of asphaltenes and weaken that of light components in asphalt.

It can be seen in Figure 10c that the diffusion rate of the SBS modifier component of S-HV modified asphalt was higher than that of PG76-22 modified asphalt, which was due to the content of SBS modifier, indicating that the increase in modifier content increased molecular movement. Among them, saturates were greatly decreased at low temperatures. The order of diffusion rate of each component of S-HV modified asphalt was negatively correlated with that of molecular weight, which was lower than that of 70# asphalt at the same temperature, especially the light component, which greatly increased its durability. This showed that SBS modifier had a great influence on the diffusion rates of light components and higher modifier contents decreased the diffusion rates of light components in asphalt.

### 3.4. Solubility Law

Since physical and chemical reactions between asphalt and C-S-H gel often take place at room temperature, in order to simulate actual situations the surface concentration variations of C-S-H gel by components of different asphalt types were studied. The solubility of each asphalt component along the X direction was simulated under equilibrium state at 25 °C.

As shown in Figure 11, the solubility of saturates in asphalt near the surface of the C-S-H gel was the highest and there were small peaks and obvious aggregations. From the solubility curve of saturates, it was seen that the order from near to far from the surface of C-S-H gel at the highest peak was as follows: 70# asphalt, PG76-22 modified asphalt, and S-HV modified asphalt, which was proportional to the content of modifier.

As can be seen in Figure 12, a great similarity was observed between the aromatics solubility curves of 70# and S-HV modified asphalts after 10 Å. The solubility of aromatics in PG76-22 modified asphalt was high at both ends and low in the middle, while this was not the case for S-HV modified asphalt. This revealed that 4.27% modifier content had a greater effect on the solubility of aromatic phenols adsorbed onto the surface of C-S-H gel than that containing 11.8% modifier content, and an appropriate amount of modifier content could regulate the adsorption of aromatics and C-S-H gel.

As can be seen in Figure 13, the solubility curves of resins were mainly concentrated in the range of 20 to 30 Å. The order of the area surrounded by the solubility curve and along the direction of the X axis in the range of 0 to 10 Å was as follows: 70# asphalt < PG76-22 modified asphalt < S-HV modified asphalt, which corresponded to the amount of modifier. Addition of modifiers improved adsorption capacity between resins and C-S-H gel.

As shown in Figure 14, the solubility of asphaltenes in PG76-22 modified asphalt had the greatest difference from those of S-HV and PG76 modified asphalts. The maximum value of asphaltenes appeared in the range of 15 Å to 20 Å and the solubility of asphaltenes at both ends was low, indicating that asphaltenes presented aggregation behavior in the middle. Relatively speaking, the distribution of asphaltenes in 70# and S-HV modified asphalts was more uniform.

## 4. Conclusions

In order to study the adsorption performance of the interface between asphalt and C-S-H gel, three types of asphalt binder and C-S-H gel models were developed by MD. The effects of different asphalt components and temperatures on the adsorption performance of asphalt-C-S-H gel interface were analyzed. The following main conclusions were drawn.

(1) Adhesion work between different types of asphalt binder and C-S-H gel was in the following order: S-HV modified asphalt > PG76-22 modified asphalt > 70# asphalt, which was consistent with SBS modifier content. Adhesion work was mainly provided by Coulomb electrostatic force and the electrostatic interaction between asphalt and C-S-H gel surface was increased by the increase in temperature.

(2) The adsorption rates of the four components of 70# asphalt on the surface of C-S-H gel was in the following order: saturates > aromatics > resins > asphaltenes, which indicates that molecular weight affected the diffusion rate to some extent. The addition of SBS modifier weakened the diffusion rate of light components in asphalt and an increase in SBS modifier content decreased the diffusion rate of light components.

(3) From the saturate concentration curves of different types of asphalt at 25 °C, it was seen that the order from near to far from the surface of C-S-H gel at the highest peak was: 70# asphalt, PG76-22 modified asphalt, S-HV modified asphalt, which was proportional to SBS modifier content. The addition of an appropriate amount of SBS modifier could adjust the adsorption of aromatics and C-S-H gel and improved the adsorption of resins and C-S-H gel.

## Figures and Tables

**Figure 1 materials-14-04406-f001:**
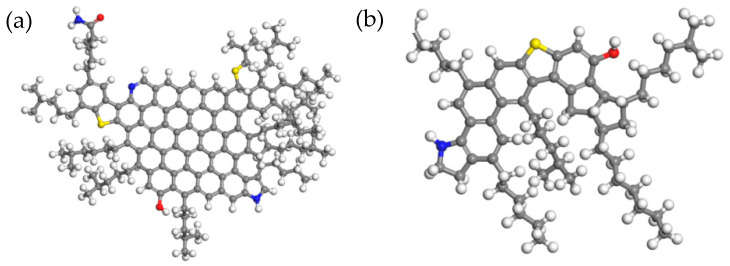
Three-dimensional structure diagram of the four components of asphalt: (**a**) asphaltenes; (**b**) resins; (**c**) saturates; and (**d**) aromatics. Note: white atoms = H, black atoms = C, yellow atoms = S, blue atoms = N, red atoms = O.

**Figure 2 materials-14-04406-f002:**
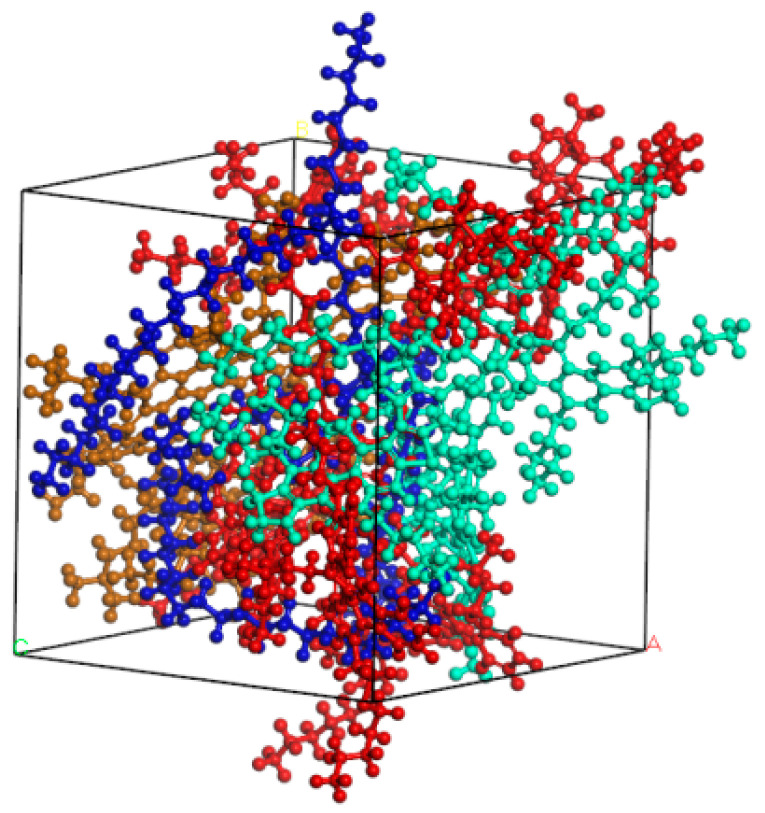
Structure of unit cell of four-component asphalt.

**Figure 3 materials-14-04406-f003:**
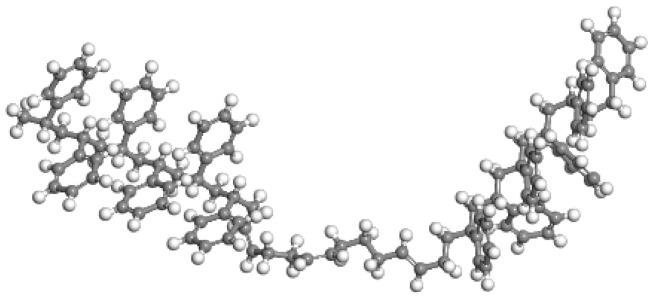
SBS modifier molecular model.

**Figure 4 materials-14-04406-f004:**
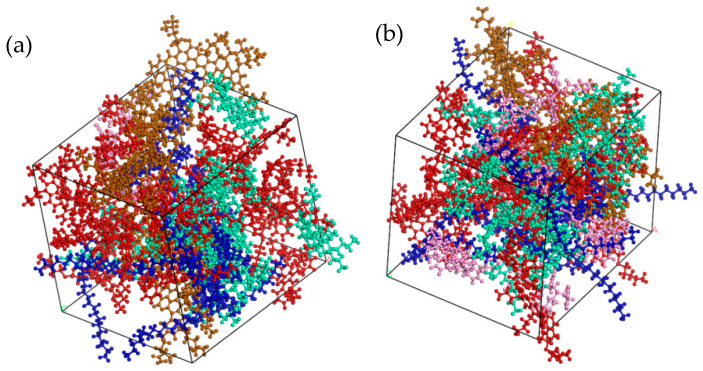
Two molecular models of modified asphalt: (**a**) PG76-22 modified asphalt (SBS modifier parameters 4.27%); (**b**) S-HV (SBS modifier parameters 11.8%).

**Figure 5 materials-14-04406-f005:**
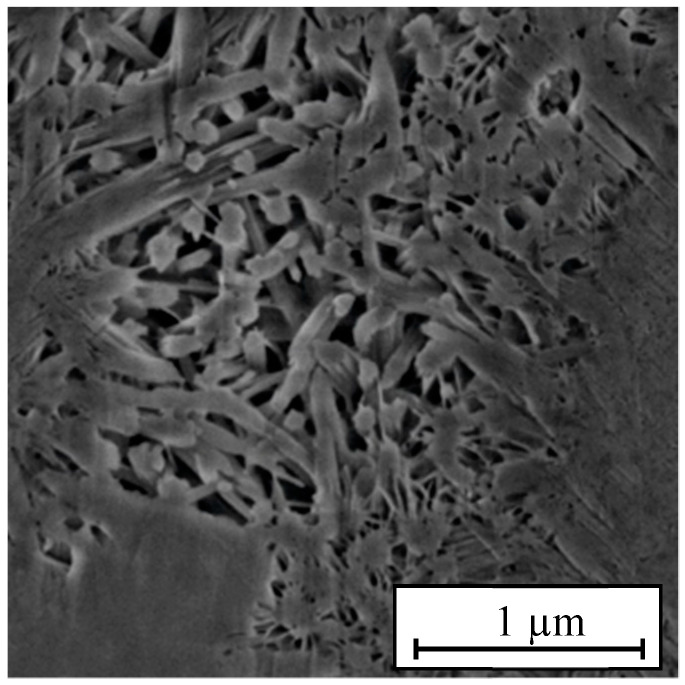
SEM image of C-S-H [37].

**Figure 6 materials-14-04406-f006:**
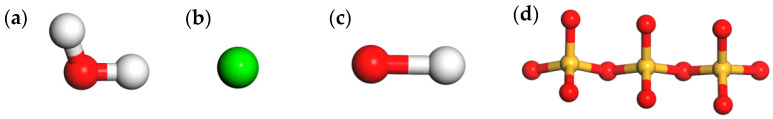
The molecular model of C-S-H gel components. (**a**) H_2_O molecular model; (**b**) Ca^+^ ion model; (**c**) OH^−^ ion model; (**d**) Si_3_O_10_ molecular model.

**Figure 7 materials-14-04406-f007:**
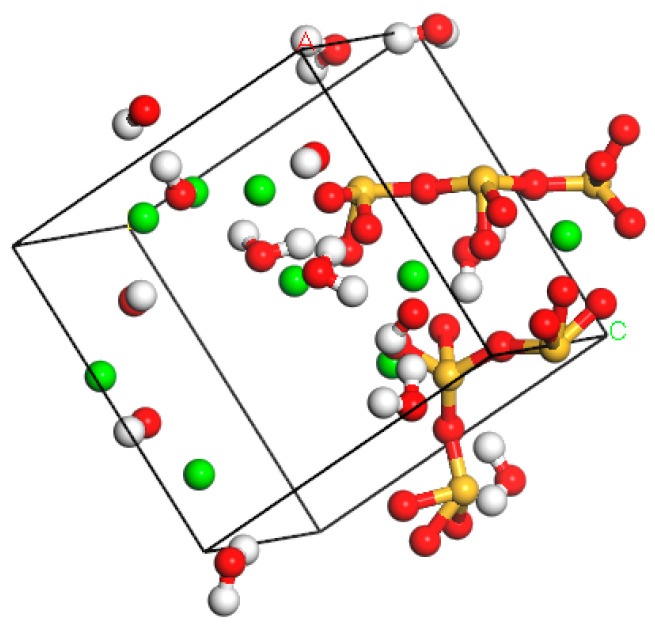
The molecular model of C-S-H gel.

**Figure 8 materials-14-04406-f008:**
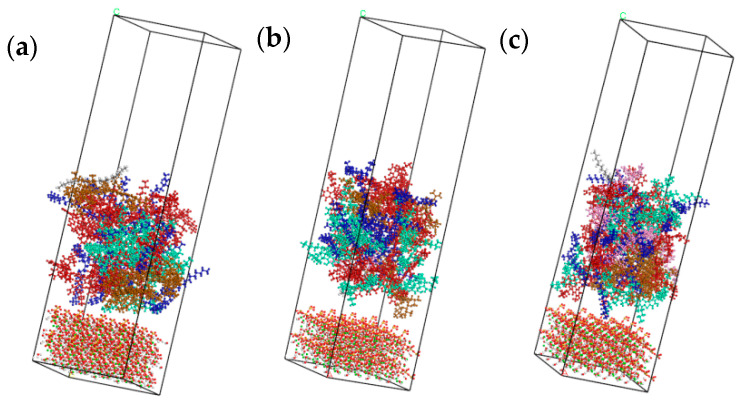
Interface adhesion models for different asphalt types. (**a**) Model A (70# asphalt); (**b**) Model B (PG76-22 modified asphalt); (**c**) Model C (S-HV modified asphalt).

**Figure 9 materials-14-04406-f009:**
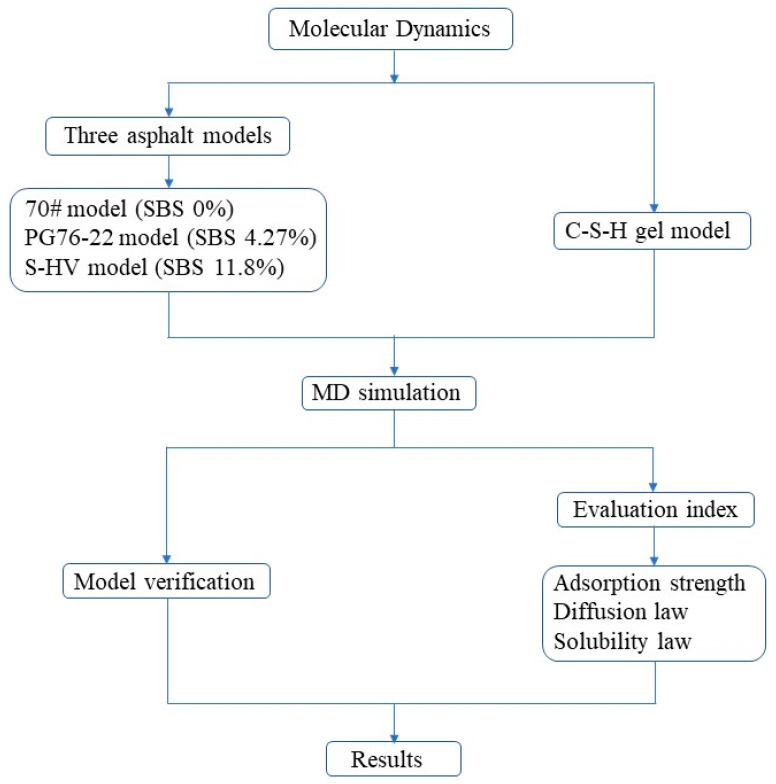
Research methods and technical workflow.

**Figure 10 materials-14-04406-f010:**
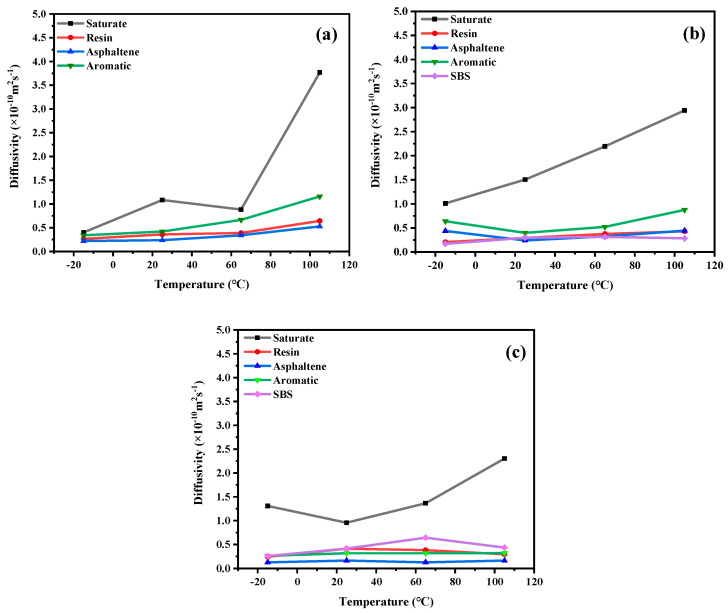
Diffusion coefficients of asphalt samples with different SBS contents at different temperatures. (**a**) 70# asphalt; (**b**) PG76-22 modified asphalt; (**c**) S-HV modified asphalt.

**Figure 11 materials-14-04406-f011:**
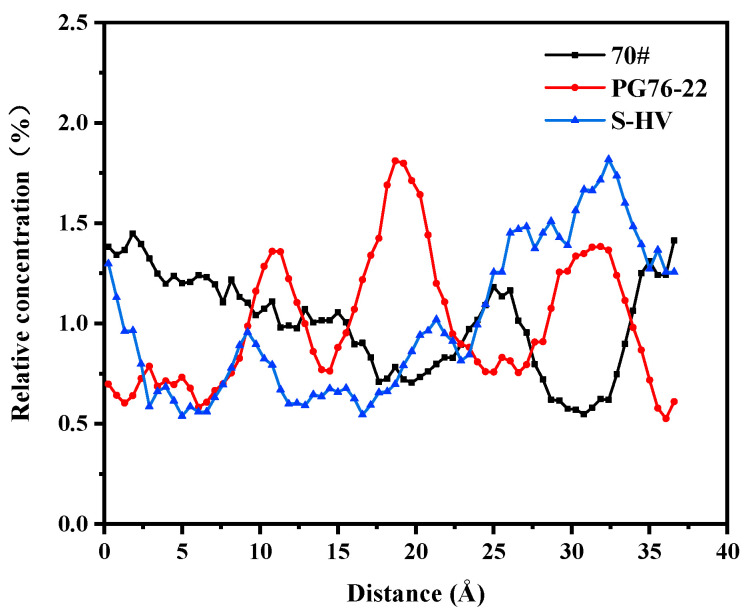
Solubility distribution of saturates.

**Figure 12 materials-14-04406-f012:**
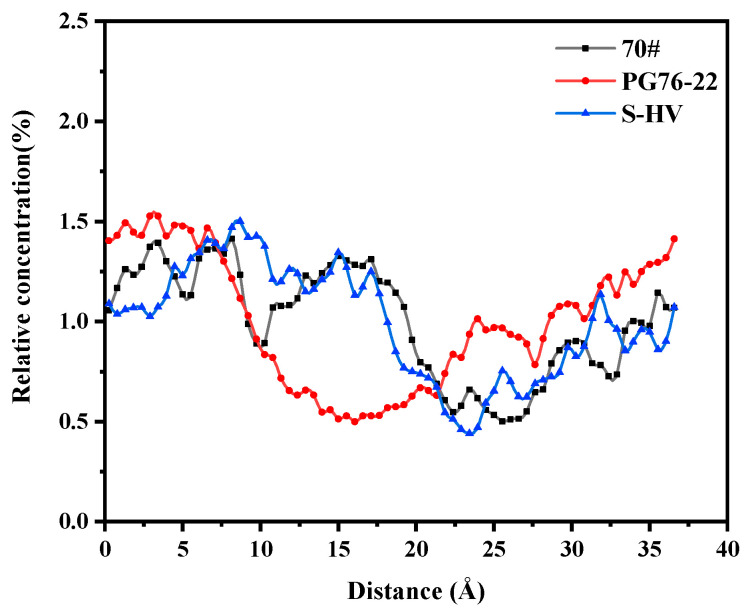
Solubility distribution of aromatics.

**Figure 13 materials-14-04406-f013:**
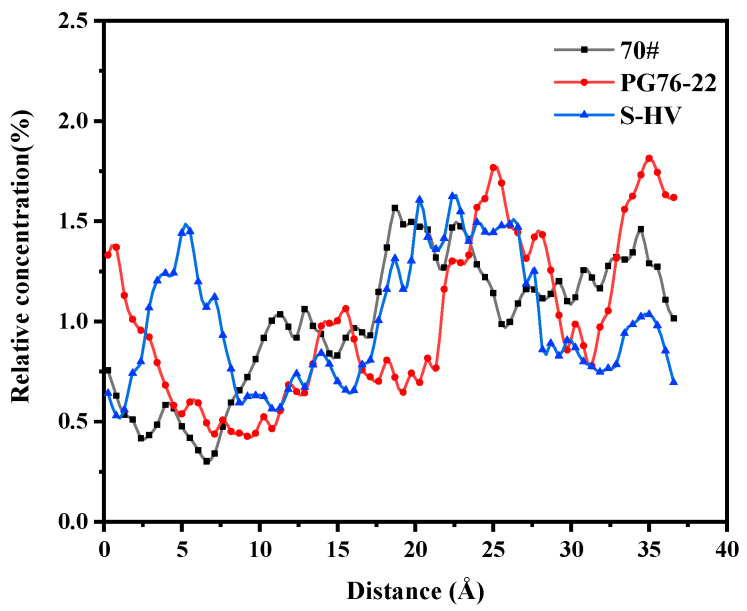
Solubility distribution of resins.

**Figure 14 materials-14-04406-f014:**
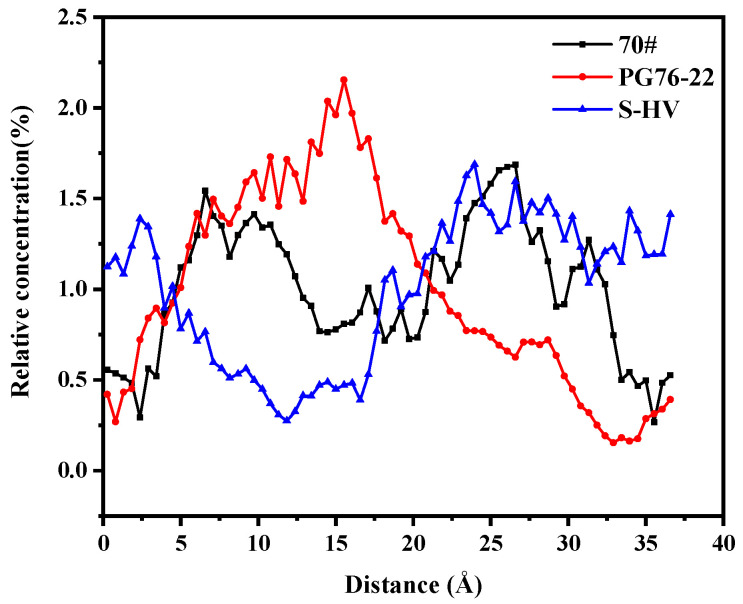
Solubility distribution of asphaltenes.

**Table 1 materials-14-04406-t001:** Detailed composition of three asphalt materials.

Chemical Fractions	70# Asphalt			PG76-22 Modified Asphalt			S-HV Modified Asphalt		
	Chemical Formula	Number of Molecules	Mass Fraction (%)	Chemical Formula	Number of Molecules	Mass Fraction (%)	Chemical Formula	Number of Molecules	Mass Fraction (%)
Asphaltenes	C_149_H_177_N_3_O_2_S_2_	1	19.74%	C_149_H_177_N_3_O_2_S_2_	1	18.89%	C_149_H_177_N_3_O_2_S_2_	1	17.41%
Resins	C_59_H_85_NOS	3	24.07%	C_59_H_85_NOS	3	23.04%	C_59_H_85_NOS	3	21.23%
Saturates	C_22_H_46_	5	14.54%	C_22_H_46_	5	13.92%	C_22_H_46_	5	12.83%
Aromatics	C_46_H_50_S	7	41.65%	C_46_H_50_S	7	39.87%	C_46_H_50_S	7	36.73
SBS				C_35_H_56_	1	4.27%	C_109_H_118_	1	11.80%

**Table 2 materials-14-04406-t002:** Proportions of C-S-H gel composition.

Name	H_2_O	Ca^+^	OH^−^	Si_3_O_10_
Crystal ratio	8	9	6	2
Mass ratio (%)	12.04	30.12	17.04	40.80

**Table 3 materials-14-04406-t003:** Comparison of simulated and experimental results of the asphalt layer.

Project	Relative Density	Solubility Parameter
Simulated value	1.00	/(J cm^−3^)^1/2^
Actual reference value	1.02 ± 0.02	22.59

**Table 4 materials-14-04406-t004:** Adhesion work between three asphalt types and C-S-H gel.

Temperature (°C)		−15	25	65	105
70#asphalt (kcal/mol)	△*E_vdw_*	61.202	72.16	57.03	63.34
	△*E_coulomb_*	9932.60	9792.88	9789.73	10,052.12
	△*E*	9993.811	9865.04	9846.76	10,115.46
PG76-22 (kcal/mol)	△*E_vdw_*	64.45	91.366	97.52	88.83
	△*E_coulomb_*	10,575.03	10,442.46	10,475.25	10,576.89
	△*E*	10,639.49	10,533.83	10,572.77	10,665.72
S-HV (kcal/mol)	△*E_vdw_*	51.98	57.68	42.86	67.13
	△*E_coulomb_*	11,352.95	11,310.60	11,418.13	11,365.12
	△*E*	11,404.93	11,368.28	11,460.99	11,432.25

## Data Availability

The data that support the findings of this study are available from the corresponding author upon reasonable request.

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
