# Peer review of "Study on the Adhesion Performance of Asphalt-Calcium Silicate Hydrate Gel Interface in Semi-Flexible Pavement Materials Based on Molecular Dynamics"

_materials, 2021, doi:10.3390/ma14164406_

Round 1
Reviewer 1 Report
This paper presents studies on the adhesion performance of asphalt- calcium silicate hydrate gel interface in semi-flexible pavement materials based on Molecular Dynamics. For this purpose, the adhesion performance of asphalt-C-S-H gel interface in semi-flexible pavements at molecular scale has been investigated. Comparisons between the results shows that the addition of SBS modifier could regulate the adsorption of aromatics by C-S-H gel in the four components of asphalt and improve the adsorption of resins by C-S-H gel.
This is an interesting paper. However, this reviewer does not recommend the publication of the manuscript in the present form because of the following reasons:
1) This reviewer think it will be useful if the authors provide some additional information on the asphalt model used in this study.
2) English usage and spelling should be improved.
3) The quality of the figures should be improved.
3) The manuscript is focused on properties of flexible pavements and needs better description of the properties of asphaltic materials in pavement design, such as linear and nonlinear viscoelastic properties. They may use available literature such as the following reference:
- (2020). Effect of Evotherm-M1 on Properties of Asphaltic Materials Used at NAPMRC Testing Facility. Journal of Testing and Evaluation, 48(3).
- (2019). Characterization and validation of the nonlinear viscoelastic-viscoplastic with hardening-relaxation constitutive relationship for asphalt mixtures. Construction and Building Materials, 216, 648-660.
- (2018). A straightforward procedure to characterize nonlinear viscoelastic response of asphalt concrete at high temperatures. Transportation Research Record, 2672(28), 481-492.
- (2018). Experimental and Analytical Procedures to Characterize Mechanical Properties of Asphalt Concrete Materials for Airfield Pavement Applications. In Civil, Environmental and Architectural Engineering, Ph.D., University of Kansas, Lawrence, KS, 2018. p. 247.
Reviewer 2 Report
Please see the attached reports.

Reviewer 3 Report
This study investigates the interface between asphalt and calcium silicate hydrate (C-S-H) gel in semi-flexible pavement materials. Some suggestions to improve the manuscript are as follows:
Line 15 is not clear what it refers to with #70
Line 31 the authors are kindly invited to review and possibly cite the following paper that deals with the pavement bearing capacity and rutting resistance: in International Journal of Pavement Engineering, 2018, 19.6: 479-488. https://doi.org/10.1080/10298436.2016.1175562
Line 34 what are the authors referring to with cement mortar? The mastics inside the paper have never been mentioned, as they are an important component in asphalt mixtures. The authors are kindly invited to review and possibly cite the paper that deals with the asphalt mastics: in Construction and Building Materials, 2021, 270, 121394. https://doi.org/10.1016/j.conbuildmat.2020.121394 and in Road Materials and Pavement Design, 2019, 20.3: 592-607. https://doi.org/10.1080/14680629.2017.1407818
Line 64 what is the S-HV modified asphalt?
It is suggested to improve the introduction section with a concise description of the work carried out
It would be useful to insert a photo of the C-S-H gel composition.
It is suggested to insert a descriptive section of the materials
Round 2
Reviewer 2 Report
Thank you for the corrections.
This manuscript is a resubmission of an earlier submission. The following is a list of the peer review reports and author responses from that submission.